# Potential of Naturally Derived Compounds in Telomerase and Telomere Modulation in Skin Senescence and Aging

**DOI:** 10.3390/ijms22126381

**Published:** 2021-06-15

**Authors:** Barbara Jacczak, Błażej Rubiś, Ewa Totoń

**Affiliations:** Department of Clinical Chemistry and Molecular Diagnostics, Poznan University of Medical Sciences, 49 Przybyszewskiego St., 60-355 Poznań, Poland; barbarajacczak07@gmail.com (B.J.); blazejr@ump.edu.pl (B.R.)

**Keywords:** telomerase, telomeres, natural compounds, senescence, skin aging, antioxidants

## Abstract

Proper functioning of cells—their ability to divide, differentiate, and regenerate—is dictated by genomic stability. The main factors contributing to this stability are the telomeric ends that cap chromosomes. Telomere biology and telomerase activity have been of interest to scientists in various medical science fields for years, including the study of both cancer and of senescence and aging. All these processes are accompanied by telomere-length modulation. Maintaining the key levels of telomerase component (hTERT) expression and telomerase activity that provide optimal telomere length as well as some nontelomeric functions represents a promising step in advanced anti-aging strategies, especially in dermocosmetics. Some known naturally derived compounds contribute significantly to telomere and telomerase metabolism. However, before they can be safely used, it is necessary to assess their mechanisms of action and potential side effects. This paper focuses on the metabolic potential of natural compounds to modulate telomerase and telomere biology and thus prevent senescence and skin aging.

## 1. Introduction

Telomeres are nucleoprotein complexes located at the ends of linear chromosomes. These structures are composed of (a) specific DNA repeat sequences of 5′ TTAGGGn (conserved in vertebrates), and (b) the shelterin complex, which consists of six proteins that maintain the structure and control telomere metabolism [1]. Their primary function is to protect the linear chromosome ends against damage which could lead to DNA breaks such as single-strand breaks (SSB) or double-strand breaks (DSB). Thus, they provide chromosome integrity and stability [1]. They also protect chromosomes from joining, which would have fatal results for the cell [1]. In most normal cells, telomeres become shorter with every cell division due to the end-replication problem [2]. This issue can be compensated by telomerase, a specialized ribonucleoprotein (RNP) complex containing two major components and several complementary proteins [2]. The main subunits of the enzyme are (a) the telomerase RNA component (TERC), which plays the role of RNA template, and (b) the catalytic element, human telomerase reverse transcriptase (hTERT). The crucial role of hTERT is to catalyze the synthesis of new telomeric repeats based on the RNA template. It is noteworthy that the expression and activity of this enzyme are observed in the early stages of prenatal development. Later, telomerase expression and activity are significantly limited in most normal cells [3].

Interestingly, hTERT has been reported to play other, telomere-independent functions. One of them is associated with the modulation of mitochondrial metabolism and free radical elimination [4]. This area is of high importance when considering the anti-aging and antisenescence potential of telomerase.

Aging is defined as an age-dependent functional decline of cells, tissues, and whole organisms with gradual losses of reproductivity [5], which is determined by multiple genetic and environmental factors. Consequently, it starts at different moments for different individuals [4], but over time, the body’s metabolic systems become less efficient and cells become more susceptible to various harmful factors. Eventually, the regenerative abilities of individual organs are diminished. An in-depth understanding of aging mechanisms will bring us closer to understanding numerous pathomechanisms of age-related diseases. Many theories suggest that various aging phenotypes appear because of the age-dependent accumulation of damage [6,7]. This may eventually lead to senescence, which is defined as irreversible cell-cycle arrest driven by a variety of mechanisms, including telomere shortening, genotoxic stress, mitogens, inflammatory cytokines, and activation of the p53 tumor suppressor and/or the cyclin-dependent kinase inhibitor p16 [8]. Cellular senescence is thought to contribute to age-related tissue and organ dysfunction [9].

Both aging and senescence may result from internal or external signaling, and both contribute to the dysfunction of metabolic pathways. On the other hand, both, by limiting the proliferation of potentially damaged cells (due to exposition to harmful conditions), protect the organism from the consequences of dissemination of damaged cells.

The most harmful factors affecting cell metabolism are UV radiation, air pollution, tobacco smoke, and a high-fat and high-carbohydrate diet. Considering the direct effect of external, physical factors, the first line of the human organism’s defense is the skin. As time goes by, skin cells’ (and most somatic cells’) proliferative potential and resistance to apoptosis are decreased, and the secretion of factors that promote inflammation and tissue decline increases [10,11,12].

## 2. Molecular Basis of Senescence and Aging

There are three major hallmarks of aging [13]. The first is the set of primary drivers that cause damage. These are damage to the telomeres, damage to DNA, and epigenetic and mitochondrial dysfunction. The second covers antagonistic drivers that act in response to the damage, of which the main factor is senescence, manifested by irreversible growth arrest of cells. Consequently, it contributes to aging and age-related diseases that are accompanied by changes in the phenotype of an organism, chromatin remodeling, metabolic changes, increased autophagy, and the release of numerous complex pro-inflammatory factors. Senescence can occur due to normal aging, due to age-related diseases, and due to therapy (such as chemotherapy). The third refers to integrative aging drivers that are a consequence of the damage acquired over a cell’s lifetime, and is represented by proteostatic dysfunction and disruptions in the signaling pathways [14].

The lifespans of all somatic cells are limited, as Hayflick and Moorhead observed in 1961 [15]. It is noteworthy that these experiments were performed with the use of cells of different origins, including skin fibroblasts. Significantly, the metabolic alterations in cells (including senescence) are highly correlated with the culture (and environmental) conditions, including energy restrictions or oxygen levels [15]. Interestingly, the reactive oxygen species (ROS) levels in the skin are the highest of all tissues. Moreover, a clear correlation exists between ROS levels and skin aging [15].

The maximal number of cell divisions is usually cell-type-dependent, but on average is approximately 50 (limited by the end-replication problem, as mentioned above) [16]. Thus, replicative senescence reflects the number of cell divisions (and proliferative and metabolic activity) rather than time. It is modulated by numerous genes and biological factors, including oxidative stress, mitochondrial dysfunction, and replicative exhaustion [13]. It is accompanied by a pro-inflammatory phenotype that includes secretory cytokines, growth factors, metalloproteinases, and reactive oxygen species (Figure 1) [17]. Importantly, cellular senescence can convert neighboring cells to senescence in a paracrine manner [18], and they can accumulate during aging. Accompanying telomere attrition can lead to critically short telomeres that are detected by the cells as double-strand breaks. DSBs trigger the DNA damage response (DDR), a signaling cascade mediated by the ataxia teleangectasia-mutated (ATM) kinase and ataxia telangiectasia and Rad3-related (ATR) kinase, which activate p53 and lead to cell-cycle arrest and senescence [19]. Specifically, DNA damage triggers phosphorylation of the protein at serine 15 (Ser-18 in mouse) [20]. Several p53 targets and regulators have been linked to the induction of senescence, including critical ones such as CDKN1A/p21 (cyclin-dependent kinase inhibitor), which was shown to be upregulated during replicative senescence [21]. This essential mediator of p53-dependent cell-cycle arrest contributes to the growth arrest of senescent cells, but also regulates numerous target genes affecting several physiological and metabolic pathways, including mTOR and autophagy, that may contribute to mitochondrial dysfunction and senescence [19]. Another key pathway that controls senescence is p16INK4A-Rb. p16, a cyclin/cdk inhibitor, prevents phosphorylation of Rb by cyclin/cdk complexes [22]. This in turn leads to inhibition of cell proliferation by inhibitory binding to E2Fs transcription factors (pro-proliferative signaling) [23]. This results in an increased accumulation of γH2A.X and 53BP1 in chromatin, which in turn activates ATM and ATR, as well as CHK1 and CHK2. Altogether, the p53-dependent pathway is induced [24], which leads to altered transcription of the cyclin-dependent kinase inhibitor p21CIP1. In turn, p21CIP1 blocks CDK4/6 activity, resulting in hypophosphorylated Rb, and cells enter a quiescent state [25]. Importantly, the sustained induction of p53 allows for a permanent arrest of the cell cycle through persistent induction of p21CIP1 [19].

## 3. Skin Structure and Telomerase

The skin is one of the largest organs in the human body. There are three layers in this tissue which are separated but cooperate to maintain homeostasis: the epidermis, dermis, and hypodermis (Figure 2) [26].

The epidermis consists of keratinocytes, Langerhans cells, melanocytes, neuroendocrine cells, and inflammatory cells [27]. The most abundant are keratinocytes. They produce and store keratin, which provides stiffness and water-resistant properties to the skin and its appendages (hair and nails). Moreover, the tight packing of keratinocytes protects the skin against pathogenic invasion. The other group of cells in the epidermis are melanocytes. They produce skin pigment, melanin, involved in the protection against terrestrial solar ultraviolet radiation (UVR) [28]. Melanocytes express MHC II elements (main histocompatibility complex) [29] and, consequently, are associated with the immune system’s functioning. The third group of cells in of the epidermis is the Merkel cells. They are located in the stratum basal, on the border of the epidermis and the dermis. Merkel cells are mechanoreceptors that communicate with cutaneous neurons in skin sensation [30]. In the epidermis, there are two representatives of bone-marrow-derived cells: Langerhans cells (LHCs) and dendritic cells (DCs). Their primary role is to trigger tolerance to commonly present and harmless antigens.

The second layer of the skin, the dermis, is responsible for nutrient and blood supply to the skin. It is separated from the epidermis by a structurally and chemically complex basement membrane zone comprising intertwining collagen fibers. Structurally, the dermis is a connective tissue composed of collagen and elastin fibers embedded within the basic substances produced by fibroblasts; it is abundant with lymph and hemocirculation, nerves, and additional structural components (hair follicles and sweat glands). The majority of dermis cells are fibroblasts, which possess multifunctional activity but are predominantly involved in the production of extracellular matrix (ECM) components [31,32]. Like keratinocytes, fibroblasts are capable of producing cytokines [30], so they also can contribute to the response to pathogenic invasion mediated by inflammation.

The innermost layer of the skin is called the hypodermis. It consists of connective tissue, mainly built from fat cells, with well-developed blood vessel network. The main structure of the hypodermis is adipose tissue, the thickness of which varies in different body locations. The hypodermis facilitates cutaneous mobility. The adipose tissue serves as metabolic energy storage and provides thermal insulation, cushioning properties, and shock absorption for the integumentary system [33,34].

Apart from being a protective barrier, the general function of the skin is to maintain homeostasis, including prevention of the percutaneous loss of electrolytes, fluid, and proteins; it also controls immune activity and sensory perception as well as regulating temperature [31,33,34]. Thus, the skin is a very complex tissue that drives complex metabolic interactions over the whole lifespan. It is noteworthy that different skin cells show different structures, different functions, and also different levels of telomerase activity and expression (Table 1). Telomerase has been detected in keratinocytes of the basal epidermis, but not in skin fibroblasts [35]. It has been shown that telomerase activity in keratinocytes can be modulated by various factors, including ultraviolet (UV) light, poison ivy, and inflammation [36], suggesting the enzyme’s protective role. Even if not all of these actions are related to telomere restoration, they can still affect cell proliferation and regenerative potential, which are crucial in aging prevention. Thus, knowing the location of telomerase in skin cells is particularly important when considering the possibility of using telomerase activity modulators as potential anti-aging strategy elements.

The natural renewal of the skin occurs through the action of skin stem cells (SSCs), i.e., skin-derived progenitor cells (SKPs) and dermal sheath cells (DSCs) [27,44]. Skin stem cells are found in the dermis, specifically in the basal layer. [44,45,46,47,48,49,50]. In both cell types, telomere stabilization is driven by hTERT [50,51,52,53,54,55]. Comparative studies of skin stem cells and hair follicle stem cells (HFSC) have revealed significantly higher hTERT expression levels in HFSCs than SSCs [41]. Similarly, in mouse hair follicles, telomerase was shown to be associated with the anagen phase, a period of intense progenitor cell activity [56]. Moreover, the function of hTERT in HFSC is not limited to protecting chromosomes from age-related shortening. The telomerase catalytic subunit was shown to be sufficient to induce proliferation of mouse hair follicle stem cells, and this function was independent of its role in telomere synthesis [42,57].

## 4. Aging in Skin

The most evident and visible symptoms of aging in humans manifest as changing skin appearance due to continuous exposure to endogenous and exogenous factors. A decrease in the regenerative potential of skin cells is often linked to a decline in the elimination of senescent cells, and their gradual accumulation results in the physiological aging of the tissue itself [44]. Skin aging is a multifactorial process that influences the structural and functional features of the skin. It is driven by intrinsic (e.g., time, genetic factors, hormones) and extrinsic (e.g., UV exposure, pollution) factors. Hence, cutaneous aging is involved in two simultaneously occurring processes: intrinsic aging and extrinsic aging. Intrinsic aging, known as chronological or natural aging, is genetically determined, whereas extrinsic aging is caused by environmental factors such as chronic sun exposure, which is known as photoaging [27]. Terrestrial solar ultraviolet radiation is the most important environmental factor affecting skin physiology.

The primary features of aging skin include changes in the skin hydration level, loss of thickness and flexibility, the appearance of wrinkles, and some aberrant pigmentation, especially on the skin of the hand. UVA induces immediate pigment darkening mediated by photo-oxidation of pre-existing melanin and redistribution of melanosomes [58], leading to hair greying and hair loss [59].

The most significant result of skin aging is the reduced proliferative capacity of fibroblasts, keratinocytes, and melanocytes [60]. Curtailment of extracellular matrix (ECM) synthesis in the dermis is caused by the increased expression of matrix-degrading enzymes and matrix metalloproteinases (MMPs), which are mainly secreted by epidermal keratinocytes and dermal fibroblasts [58]. MMPs lead to the degradation and accumulation of a nonfunctional matrix due to cross-links in collagen fibers (intrinsic aging) or partially degraded elastin fibers (extrinsic aging) in combination with increased oxidative stress and inflammatory process [12].

## 5. Roles of Telomeres in Skin Aging

The primary indicators of aging are telomere attrition, epigenetic and mitochondrial dysfunction, and general DNA damage. Significantly, this process can be accelerated by stress, which leads to the accumulation of damage and eventually to mitotic crisis or cell death [61]. Numerous agents can trigger DNA damage, including ionizing radiation, radiomimetic drugs, reactive oxygen species, metabolic errors during replication and transcription [62], and deficient DNA repair [63]. Some of those factors are just a result of cells’ metabolic activity and cannot be eliminated, but some of them can be prevented to slow down senescence and aging.

Evaluation of the aging of skin has revealed the primary roles of alterations observed in the metabolism of fibroblasts. In young skin, fibroblasts adhere to the surrounding intact extracellular matrix (ECM), which is mainly comprised of type I collagen [64,65]. In aged skin, fibroblast attachment is impaired due to progressive ECM degradation, resulting in fibroblast size reduction, decreased elongation, and collapsed morphology [64,66,67,68,69]. Simultaneously, their proliferative potential is also significantly reduced, which impedes skin regeneration, thus contributing to progressive aging. The reduction of dermal fibroblast spreading and cell size can also increase mitochondrial ROS generation, which induces senescence [70].

### 5.1. Oxidative Stress

Due to the high G content in telomeric structures, the telomere sequence is sensitive to oxidative stress [71], alkylation [72], and ultraviolet (UV) irradiation [73], which are the leading causes of skin aging. It has been shown that telomere length in human dermal fibroblasts can be shortened by a single high dose of UVA radiation, and that acute photodamage might contribute to early photoaging in human skin via rapid telomere shortening [74]. A recent study demonstrated that 8-oxoG, a typical oxidative DNA lesion, was correlated with replicative stress and subsequently loss of telomeric repeats. It was also accompanied by reduced cell proliferation, indicating that oxidative stress is a significant contributor to telomere dysfunction [75]. Furthermore, damage is less efficiently repaired in telomeres compared to the bulk of the genome, mainly because of the shelterin complex elements that limit the access of the DNA repair machinery to chromosome ends as a way to prevent telomere fusions [76]. Moreover, it turned out that the expression of telomerase (hTERT) might be insufficient to rescue stress-induced telomere dysfunction and prevent senescence [77,78].

ROS can be produced from different sources, including the mitochondrial electron transport chain, peroxisome and endoplasmic reticulum (ER) proteins, the Fenton reaction or cyclooxygenases, lipoxygenases, xanthine oxidases, and nicotinamide adenine dinucleotide phosphate (NADPH) oxidases [79]. Oxidation products and ROS can cause the activation of receptor tyrosine kinases (RTKs). This, in turn, provokes the activation of nuclear factor, κB (NF-κB), and transcription factor activator protein 1 (AP—1). They repress collagen production and increase *MMP gene* transcription, resulting in the decreased collagen content in photoaged skin [80]. Accordingly, the use of antioxidants in an anti-aging strategy has its scientific justification: reduction of the amount of ROS, which are natural products of cellular metabolism and which contribute to actual damage to cellular and tissue structures. Surprisingly, it was shown in a previous study that cells with longer telomeres might be more sensitive to oxidative stress [81]. It was suggested that these structures are particularly vulnerable to oxidative damage, and thus whole cells become more susceptible to oxidative stress and its consequences.

### 5.2. Inflammaging

One of the most characteristic processes that occurs in the aging body is a slow-developing, progressive inflammatory process. The phenomenon of inflammation associated with cellular aging is called “inflammaging”. The term was first proposed in 2000 by Claudio Franceschi [82]. It plays a role in the initiation and progression of age-related diseases such as type II diabetes, Alzheimer’s disease, cardiovascular disease, frailty, sarcopenia, osteoporosis, and skin aging [83]. One of the critical drivers of inflammaging development in the skin is UV radiation, the main culprit of oxidative stress in the skin epithelium. Damaged cellular structures and products of lipid oxidation are identified by the immune system as abnormal and thus threatening to the proper functioning of the body. The complement system induces phagocytosis and activates macrophages, which have the task of removing damaged cells and lipid oxidation products [84,85]. The effect of macrophage activation is the release of MMPs involved in the damage of the extracellular matrix and the connections between the epidermis layer and the dermis. Overburdened macrophages release pro-inflammatory cytokines and ROS [86,87,88], the former of which cause chronic inflammation and long-term damage to the dermis [89].

Moreover, it was shown in one study that human skin fibroblasts from old donors can secrete IL6 and thus contribute to a chronic low-grade inflammatory state [90]. A negative relationship between telomere length and IL-6 levels was observed in a large-scale study of 1962 individuals [91]. Similarly, a negative correlation was observed between telomere length and C-reactive protein (CRP) [84]. On the other hand, increased peripheral blood expression levels of IL-6 and TNF-α in patients with metabolic disorders were shown to correlate with elevated levels of telomerase activity [92].

## 6. Nontelomeric Functions of Telomerase

Numerous studies have shown that telomere length and telomerase activity or the critical subunit hTERT play essential roles in stress-mediated aging [93]. Consequently, an anti-aging potential of hTERT in the context of telomere-independent functions of telomerase has also been postulated [94]. Recent studies have reported a contribution of hTERT to a broad spectrum of pathways, including signal transduction, gene expression regulation, and mitochondrial function, with consequences for the control of cell survival, proliferation, differentiation, migration, and regeneration [95,96,97,98,99,100]. As demonstrated by Masutomi et al., downregulation of hTERT reduced the level of H2AX phosphorylation (DSB marker) and ATM autophosphorylation (DSB repair pathway) in fibroblasts exposed to DNA-damaging agents while telomere length was unchanged [101]. Other articles have reported a protective role of hTERT in oxidative stress via the reduction of reactive oxygen species levels [102,103,104,105,106,107]. The antioxidative role of hTERT is associated with the cytoplasmic and mitochondrial localization of this subunit [108,109]. The protective role of hTERT is multidirectional. Mitochondrial hTERT has been shown to increase membrane potential, reduce ROS levels, protect mitochondrial DNA, and inhibit induction of apoptosis [110,111,112,113]. Additionally, hTERT is involved in the induction of the expression of several genes (e.g., IL6, IL8, and tumor necrosis factor-α (TNFα)) whose transcription is controlled by the NF-κB pathway [114]. Similarly, hTERT controls the expression of metalloproteinases (MMPs) via NF-κB-dependent regulated transcription [106]. Since NF-κB is a crucial regulator of inflammation, hTERT is indirectly involved in regulating this process [115].

The noncanonical function of telomerase has also been observed in skin stem cells. In 2005, it was reported that hTERT overexpression led to increased epidermal stem cell maintenance and proliferative potential independently of its telomere-lengthening function [116,117]. Thus, hTERT was recognized to act as a transcription (co-)factor to regulate gene expression [118]. hTERT was also shown to interact with the chromatin remodeling factor BRG1 [119]. This ATPase regulates the expression of selected genes and is required for the β-catenin-mediated transcription of target genes [120]. Both proteins stimulate the β-catenin-mediated pathway and promote the stem-cell self-renewal program, proliferation, and survival [119]. In turn, the β-catenin signaling pathway has been reported to be involved in the transcriptional regulation of hTERT and hair growth regulation [121]. Specifically, hTERT overexpression induced the proliferation of stem cells in the hair follicle bulge region, resulting in histological changes around the hair follicle and subsequent hair growth [122,123].

The discovery of the additional role of hTERT as an antioxidant factor has revealed new therapeutic possibilities in anti-aging strategies. To date, telomeres and telomerase have been identified as the main targets of free radical attack, which has been associated with cells’ genomic instability and leads to acceleration of the aging process in the body. Thus, the telomerase catalytic subunit, hTERT, might be one of the most promising targets for anti-aging prevention strategies (Figure 3).

In conclusion, the role of telomerase, and especially the hTERT subunit, in skin aging is not limited to protecting telomeres from shortening. The catalytic subunit of telomerase has various regulatory properties in oxidative, inflammatory, gene expression, and cell proliferation processes. It seems to help the cells to adapt to stressful situations caused by external (UVR) and internal factors (ROS). The multidirectional action of telomerase occurs in the cell nucleus, mitochondria, and cytoplasm. Consequently, any disturbance in telomerase expression or activity can result in cellular disorders, leading to induced aging associated with critical telomere length and mitochondrial dysfunction.

## 7. Telomerase Restoration and Potential Threats

Out of the two key telomerase subunits, hTR is constitutively expressed in all tissues regardless of telomerase activity [124]. It is noteworthy that it shows fivefold higher expression in cancer cells than in normal ones [125]. In contrast, hTERT is generally repressed in normal cells and upregulated in cancer cells, suggesting its primary role in enzyme activity and cancer characteristics [126]. The main mechanism controlling hTERT expression is based on transcriptional modulation [127] that reflects the pool of transcription factors available in the tissue, as well as the chromatin structure organization controlled by epigenetic regulation. The hTERT promoter is unmethylated in normal cells, and methylation is required for hTERT expression [128]. Similarly, histone acetylation/deacetylation has been shown to be involved in hTERT transactivation/repression in human cells [120]. Additionally, certain viruses encode proteins that induce hTERT transcription, including Epstein–Barr virus (EBV), cytomegalovirus (CMV), Kaposi sarcoma-associated herpesvirus (KSHV), human papillomavirus (HPV), hepatitis B virus (HBV), hepatitis C virus (HCV), human T-cell leukemia virus-1 (HTLV-1), and others [129]. In this case, the targeted induction of TERT expression is one of the key mechanisms of virus-mediated carcinogenesis. Thus, attention should be paid to strategies based on telomerase restoration and their potential negative, unexpected side effects, even if most studies report that restoration of telomerase activity in cancer cells is secondary to carcinogenesis. This is supported by the observation that telomeres in cancer cells are significantly shorter than in normal cells, which results from the fact that carcinogenesis is preceded by intensive replication, followed by the crisis and, finally, restoration of hTERT expression [130].

## 8. Telomerase- and Telomere-Based Anti-Aging Strategies in Skin

The basic mechanisms of aging and senescence are known, and the potential role of telomerase and telomeres in those mechanisms is recognized. Thus, a new perspective for anti-aging strategies has opened.

### 8.1. Telomeres and Telomerase Modulation—Food

It seems that providing a balanced diet might be the easiest way to obtain the desired anti-aging therapeutic effect, especially when combined with physical exercise. Thus, we have reviewed herein potential telomerase modulators of natural origin that significantly contribute to the regulation of molecular mechanisms of aging and could prevent this process. In recent years, numerous reports have shown a direct correlation between healthy eating habits and skin aging [131]. Some scientists believe that particular culinary herbs and spices, such as cinnamon, cloves, oregano, and allspice, can inhibit fructose-induced glycation [132]. Some compounds, including ginger, garlic, a-lipoic acid, carnitine, taurine, carnosine, flavonoids (e.g., green tea catechins), benfotiamine, a-tocopherol, niacinamide, pyridoxal, sodium selenite, selenium yeast, riboflavin, zinc, and manganese, show some anti-aging potential [133,134,135]. Moreover, some of these compounds show the potential to modulate telomerase, i.e., polyphenols, alkaloids, triterpenes, xanthones, sesquiterpene, and more [136]. Consequently, they may be perceived as potential anti-aging agents.

Since telomere metabolism is correlated with oxidation status, one of the most important groups of nutrients of the human diet in the context of aging is the antioxidants. These compounds, together with anti-inflammatory agents, are supposed to slow down telomere attrition during aging [137]. Surprisingly, vitamin E has a broad range of beneficial activities, including antioxidative, antitumor, antidiabetic, anti-inflammatory, cardioprotective, and neuroprotective properties [138], and was reported to suppress telomerase activity in human colorectal adenocarcinoma cells. This effect is mainly due to the downregulation of *hTERT* and *c-myc* mRNA through PKC inhibition. Clearly, there are two totally different areas that must be evaluated: telomerase expression and telomere length. Both seem to correlate with two almost unrelated mechanisms (telomere-dependent and telomere-independent) that might affect senescence and resistance to stress. Surprisingly, they are not associated with the antioxidative potential of vitamin E.

The key vitamin that contributes to embryonic development, cell growth and differentiation, organ formation, immune function, and vision is vitamin A (retinol) [139]. Interestingly, Sharma et al. [140] reported that retinoic acid (RA; a vitamin A metabolite) decreased telomerase activity in cancer cells. However, low concentrations of RA (1nM) induced telomerase activity and decreased p16^INK4A^ expression, which triggered an extended lifespan in normal human oral keratinocytes [141]. Thus, it seems that the addition of this vitamin to food or skincare preparations might positively affect skin condition even if telomerase is not present in normal cells. Another critical food nutrient, vitamin D (made in the skin, essential for bone remodeling, immunity, insulin secretion, and blood pressure regulation [142]) was shown to both downregulate telomerase transcription modulation (VDR acts as a mediator) and also reduce telomerase activity [140]. Thus, it is postulated that its anti-aging effect is driven via antioxidant and anti-inflammatory properties, which might be associated with telomere stabilization [143].

Another example of a telomerase modulator is genistein, which is found in soybeans and soybean-enriched products. The biological effect of this compound is observed in the pharmacological concentration range (10–100 μM). It is capable of inducing telomerase expression and activity [144] However, one study reported that in a transgenic mouse model of prostate cancer, genistein (250 mg/kg diet) provoked increased prostate weight as compared with the control group. This study suggested possible negative side effects on patients with prostate cancer [145].

It is noteworthy that most of these studies have been performed in cancer cells that show telomerase expression and activity, which makes them a perfect cancer model, but not a perfect model for normal, mostly telomerase-negative, cells. However, skin cells, due to their telomerase-positive profile, seem to be an exceptional group of normal cells that may be targeted with telomerase modulation. Surprisingly, inhibition of oxidative stress protects telomerase activity in normal cells but causes telomerase inhibition in tumor cells. This results from different characteristics of the two cell types i.e., higher redox homeostasis threshold in cancer and higher demand for reactive oxygen species in these cells (ROS). This, in turn, offers a perspective for a specific interaction with the metabolism of normal cells.

### 8.2. Polyphenols

One of the most common skin-aging theories is the theory of free radicals. Therefore, the presence of antioxidants is an essential feature in both diets and cosmetics. Some of the most popular antioxidants used in cosmetology are polyphenols. Polyphenols are secondary metabolites of plants. In recent years, tea polyphenols, curcumin, flavonoids, silymarin, and grape resveratrol have been the most studied of the polyphenols with anti-aging properties [146]. The antioxidant action of polyphenols is related to attenuated collagen degradation, which accompanies inflammation due to the release of matrix metalloproteinases, cytokines, and specific signaling pathways (e.g., Nrf2, NF-κB, MAPK, etc.) [147,148]. In vivo studies examining the effects of polyphenol supplementation from *Spatholobus suberectus* stem on human skin epidermal keratinocytes showed that these compounds reduced ROS production and blocked UV-B-induced MAPKs phosphorylation [149], which counteracted the photo-aging process. The antioxidant and anti-inflammatory effects of polyphenols affect extracellular components, i.e., collagen and intracellular components. The reduction of inflammation significantly protects both cell membranes and the genetic material of the cell from the harmful effects of free radicals. Moreover, polyphenols support the action of endogenous hTERT, which also shows an antioxidant effect: it prevents damage to mitochondrial membranes by ROS and inhibits the accumulation of damage to genetic material induced by stress. In this context, polyphenols, although they do not directly modulate telomere length or telomerase expression/activity, exhibit a similar effect to hTERT. On the other hand, polyphenols (epigallocatechingallate) were reported to inhibit telomerase activity [150]. However, the observed metabolic (anti-aging) effect may constitute another example of telomerase’s noncanonical function.

A positive effect of polyphenols (resveratrol and fisetin) on normal skin cells was shown in an animal model. After a 35 day treatment, enhanced hair growth was noted among C57BL/6 mice [151]. Studies have shown that polyphenols augment mTERT and *β-catenin gene* expression levels in the dorsal skin cells of mice as well as in the human keratinocyte cell line HaCaT [152]. Considering the fact that hTERT is only selectively expressed in skin cells (see Table 2), naturally derived compounds might also selectively but efficiently substitute or supplement the hTERT-mediated anti-aging mechanisms.

Additionally, naturally derived compounds are capable of interacting with telomeric 3D structures, i.e., G4 quadruplexes (due to G-rich telomere content). Such G4 structures are the subjects of dynamic DNA modifications, which can affect genomic stability and integrity and alter gene expression. It is proposed that dietary nutrients, such as folate and antioxidants (see Table 3), can stabilize G4 structures and might play a beneficial role in reducing G4-induced DNA damage through changes in G4 structural stability [153]. Antioxidants may affect G4 structures via their role in preventing oxidative damage at guanine bases. Although the G4-binding compounds have been shown to stabilize telomeric DNA in vitro, it is not known whether their interactions with G4 structures occur under physiological conditions. Bioavailability also presents issues in studying these interactions.

### 8.3. Fatty Acids

The skin’s lipid layer is an essential component of the organ because it is a protective barrier, protecting against water loss and valuable minerals and the penetration of pathogenic microorganisms. The presence of unsaturated fatty acids in the diet has a tangible impact on the condition of the skin lipid barrier. Insufficient intake of essential fatty acids or abnormal fat metabolism leads to severe skin diseases [156,157]. Unsaturated omega-3 fatty acids, present in fish oil, and omega-6, which are found in vegetable oils, play a significant role in maintaining the proper redox balance in cells. Fermented fish oil (FFO) has been shown to reduce ROS production and inhibit MMPs [158]. Vegetable fats also have a beneficial effect on skin conditions. The fatty acids extracted from *Withania somnifera* seeds have good anti-inflammatory effects and exert an enhanced effect on psoriasis by reducing the release of pro-inflammatory factors (TNF-α and IL–6) [159]. The anti-inflammatory effect of this oil contributes to the inhibition of cytokine-induced cell metabolic activity, the reactivity of which is associated with the production of reactive oxygen species. Particular attention is paid to the fact that the content and amount of fatty acids in skin cells are subject to change. Skin aging may influence epidermal lipids and free fatty acid composition, and their physiological functions may be involved in the aging process [160]. From a broader perspective, some studies have revealed that fatty acids, sphingolipids, and glycerolipids are involved in the initiation and maintenance of senescence and its associated inflammatory components [161]. An interesting randomized controlled trial over 4 months showed that as the ratio between omega-6 and omega-3 in plasma decreases, telomere length increases. At the same time, telomerase activity was unaltered but a significant negative correlation between telomere length and biomarkers of oxidative stress and inflammation was reported. Consequently, it seems that supplementation with higher n-6: n-3 polyunsaturated fatty acid ratios would be beneficial [162].

### 8.4. Polysaccharides

Polysaccharides found in food can exert multidirectional effects. Their pharmacological potential includes improving immune function; antitumor, anti-virus, antiglucose, and antioxidant activities; lowering blood lipids; and low cytotoxicity. Polysaccharides are therefore ideal functional foods and active therapeutic compounds [163]. Studies have shown that polysaccharides found in marine algae (such as *Hizikia fusiforme*) significantly reduce cell ROS levels. Additionally, they prevented photo-aging by regulating the NF-κB, ap-1, and MAPKs signaling pathways [164]. Strategies to regulate aging-associated signaling pathways are expected to effectively delay and prevent age-related disorders. With considerable antioxidant and anti-inflammation capacities, herbal polysaccharides have shown some beneficial potential in preventing aging and age-related neurodegenerative diseases. Polysaccharides capable of reducing cellular senescence and modulating life span via telomere- and insulin-dependent pathways have also been found to have the potential to inhibit protein aggregation and aggregation-associated neurodegeneration [165]. Additionally, there are also some examples of polysaccharides that directly affect telomeres and telomerase (see Table 4), e.g., polysaccharides from *Cistanche deserticola*. These were shown to increase telomerase activity in the heart and brain tissues of aged mice [166]. The polysaccharides from *Astragalus membranaceus*, a well-known traditional Chinese herb, were able to increase TERT gene expression and inhibit zebrafish cell apoptosis and senescence [167]. However, some polysaccharides have also been reported to impair the telomere region or inhibit telomerase activity to exert an antitumor effect [168], but their action depends on specific disease contexts [167,169,170].

### 8.5. Keeping the Balance

Naturally derived compounds have both direct and indirect effects on telomerase activity, resulting in impacts on telomere length and stability. Most of them reveal antioxidant properties mediated by induced SOD activity (steroidal glycoside or PUFA) [185,186], reduced hydrogen peroxide, and impacts on other free radical levels (polysaccharide from *Cistanche deserticola*) [176], which indirectly protect telomere from the harmful effects of ROS. Some of them affect telomerase expression by increasing (*ginsenoside Rg1*) [154,155] or decreasing (*Epigallocatechin gallatethe*) hTERT [192]. It is noteworthy that the final effect of those compounds is dose-dependent. A perfect example of such an effect is genistein, where high concentrations block telomerase activity, while low concentrations induce this enzyme activity. It should be noted that a properly balanced diet has products belonging to different groups and with multidirectional effects. The Mediterranean diet has been proven to be one of the healthiest diets in the world due to its anti-aging effects via reduction of oxidative stress. [194,195]. The main antioxidants in the Mediterranean diet are omega-3 and resveratrol [196,197]. The effectiveness of this diet has been evidenced by studies carried out by Nurses’ Health Study, which showed that the use of the Mediterranean diet makes it possible to maintain the length of telomeres and stimulate telomerase activity in peripheral blood mononuclear cells [198,199]. However, not much is known about its effect in normal skin cells, with some reports showing the ability of resveratrol to inhibit proliferation of normal human keratinocytes in vitro [199] but also to increase hTERT expression in the human keratinocyte cell line HaCaT [198,199].

## 9. Conclusions

An in-depth understanding of the mechanisms of aging and senescence is one of the key elements of cell biology research. The answers to the questions of how the aging process is initiated, what its stages are, and their consequences will enable the design of strategies that can slow down these processes. Some of the critical elements protecting cells from the inevitable aging process are telomere length, telomerase expression, and homeostasis of the redox system. Modification of these factors using compounds of natural origin is an attractive alternative to the still widely used synthetic chemicals. Among the substances of plant origin are both inhibitors and activators of telomerase, and antioxidants. The binary action of natural compounds makes it possible to use telomerase modulators in various medical fields: oncology, where telomerase inhibitors are desired, and in dermatology, for which telomerase activators are valuable due to their regenerative potential. Significantly, the noncanonical functions of the critical telomerase subunit are associated with antioxidant activity. This may constitute the basis of a very interesting approach in regenerative medicine and cosmetics. These data may also contribute to the development of more effective anti-aging strategies, which are also important for the prevention of age-related diseases that are closely related to cellular aging. The option of using naturally derived telomerase and telomere modulators gives a comprehensive perspective in the context of improved quality and length of human life.

## Figures and Tables

**Figure 1 ijms-22-06381-f001:**
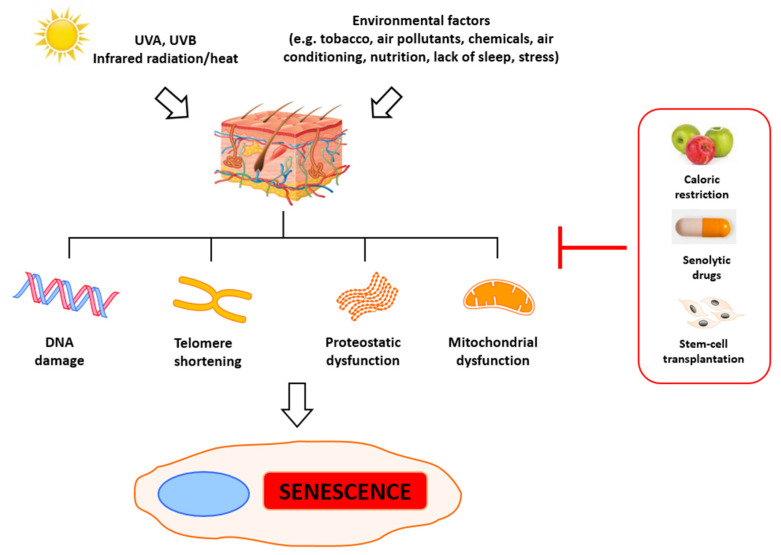
Key senescence drivers. Physical, chemical, and biological factors (e.g., tobacco, air pollution, chemicals, air conditioning, nutrition, sleep deprivation, stress, heat, UVA, and UVB) induce DNA damage, telomere erosion, oxidative stress, and proteostatic dysfunction, and consequently lead to cell senescence. Caloric restriction, senolytic drugs, and stem-cell transplantation constitute promising antisenescence strategies.

**Figure 2 ijms-22-06381-f002:**
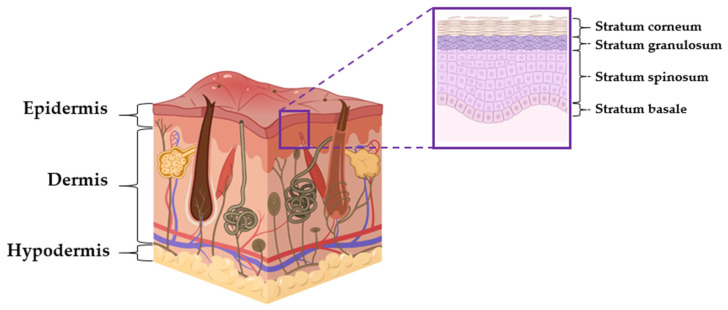
The layers of human skin. The general structure of the skin involves the epidermis (telomerase positive), dermis (telomerase low/negative), and hypodermis (telomerase low/negative). (Figure obtained from https://biorender.com).

**Figure 3 ijms-22-06381-f003:**
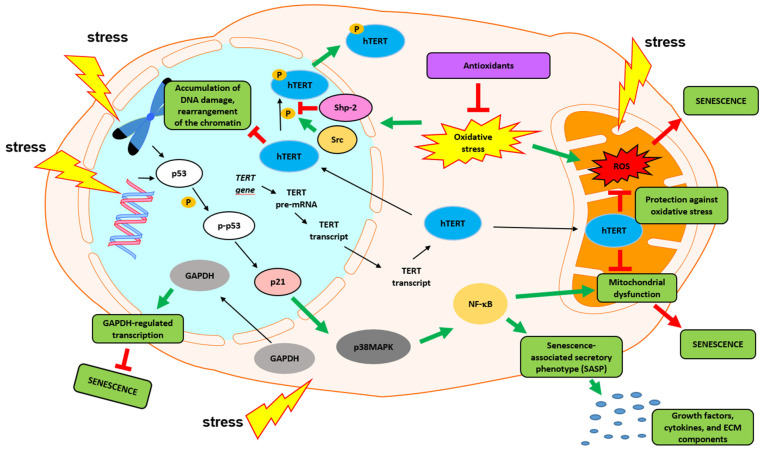
Contribution of hTERT to metabolic pathways under stress. The senescence program can be activated by different stress stimuli, such as infrared radiation, UVA, UVB, heat, chemotherapeutic drugs, replicative stress, and environmental factors, e.g., tobacco, air pollutants, chemicals, and nutrition. The main molecular effects triggered by this process are anti-apoptotic response, cell-cycle arrest, metabolic changes in mitochondria, accumulation of DNA damage, and rearrangement of the chromatin. In response to changes in the nucleus, most of the senescent cells induce the p53/p21/p38MAPK/NF-κB signaling pathway. Consequently, senescence-associated secretory phenotype (SASP) is activated and numerous growth factors, cytokines, and ECM components (e.g., EGF, IL1, IL6, MMP2, MMP3) are secreted. All of these elements play an important role in autocrine and/or paracrine signaling. Under stress conditions, hTERT is distributed between the nucleus, the cytoplasm, and the mitochondria, and plays protective roles in these organelles. In the nucleus, hTERT is required to maintain telomeres and genomic stability. It can also affect chromatin structure and modulate the DNA damage response. hTERT protects mitochondria from oxidative stress (and DNA damage) by decreasing ROS levels and binding to mitochondrial DNA (mtDNA).

**Table 1 ijms-22-06381-t001:** Expression of hTERT/telomerase in skin cells. Different cells in skin can respond to harmful factors differently also due to diverse basal hTERT expression.

Skin Cells	Relative hTERT/Telomerase Expression	Reference
**Epidermis**		
Keratinocytes	**+**	[35]
Melanocytes	**+**	[37]
Langerhan cells	**+**	[38]
**Dermis**		
Mast cells	low	[39]
Fibroblasts	-	[35]
Dermal stem cells	low	[40]
Hair follicle stem cells	+	[41]
Bulge component of the hair follicle	low	[42]
**Hypodermis**		
Fat cells	low	[43]

**Table 2 ijms-22-06381-t002:** Localization of telomerase/hTERT and its functions in cells [123].

Nuclear	Cytoplasmic	Mitochondrial
Maintenance of telomeres and genomic stability	Interaction with signaling pathways	Decrease of mitochondrial ROS and protection from stress
Interaction with signaling pathways	Redox balancing and cell adaptation to stress	Decrease of apoptosis
Regulation of chromatin structure, gene expression and modulation of DNA damage response	Telomerase complex maturation	Binding to mtDNA and protection against mtDNA damage

**Table 3 ijms-22-06381-t003:** Naturally derived compounds and their protective contributions to oxidative stress and aging.

Active Ingredients/Source	Mechanism	Reference
**Polyphenols** *tea* *curcumin* *red grapes*	Inhibition of collagen degradation by blocking the development of inflammation	[131,154,155]
**Flavonoids: catechins** *green tea*	Inhibition of AGE formation	[147,148,149]
**Herbs and spices** *ginger* *garlic* *cinnamon* *cloves* *oregano* *allspice*	Inhibition of fructose—induces glycation	[136]

**Table 4 ijms-22-06381-t004:** Mechanisms of the anti-aging effects of the active ingredients of Ayurvedic medicine and traditional Chinese medicine in the context of telomerase and telomere biology.

Active Ingredients/Source	Mechanism	Experimental Model	Reference
Polysaccharide *Cistanche deserticola* *Cinamorium songarium* *Astragalus membranaeus*	Increased telomerase activity by reducing free radicals Increased telomerase activity in testicles Telomerase activity and telomere-binding protein modification Reduction of shortening rate of telomere restriction fragment (TRF)	subacute aging model mice aging mouse model human embryonic lung diploid fibroblasts	[167] [171] [172] [173]
Pine pollen *Pinus massoniana*	Modulation of telomerase activity, increased cell population	human embryonic lung fibroblasts	[174]
Flavonoid *Euphorbia humifusa* Willd.	Regulation of telomerase activity via antioxidant effect (enhanced SOD activity)	aging mouse model	[175]
Acteoside *Cistanche tubulosa*	Increased telomerase activity, antioxidant function	aging mouse model	[176]
Astragaloside *Astragalus membranaeus*	Increased telomerase activity	human embryonic lung diploid fibroblasts	[177]
Steroidal glycoside *Cynanchum bungei*	Increased telomerase activity, antioxidant protection via the increase of SOD activity	aging mouse model	[178]
Ginsenoside Rg1 *Panax ginseng*	Decreased of telomere shortening via increased telomerase expression and restored telomerase activity	hemopoietic stem-cell ageing in mice	[154,155]
Allicin *Allium sativum* Linn.	Restored telomerase activity	fibroblast cells	[179]
Triterpenoid saponins *Centella asiatica* (L.) Urban	Nine-fold increase of telomerase activity, inhibition of the negative effects of H_2_O_2_ on DNA	peripheral blood mononuclear cells	[180,181]
Withanolide *Withania somnifera* (L.) Dunal	Increased telomerase activity Decreased effects of H_2_O_2_-induced damage on DNA	human HeLa cell	[182,183]
Basil oil *Ocimum basilicum* L.	Downregulation of the telomeric repeat binding factor 1 (TERF–1), which is a telomere length suppressor	K562 cells (chronic myelogenous leukemia)	[184]
Polyunsaturated fatty acids (PUFA) *11,14,17*—*eicosapentaenoic acid (ETA)*	Suppression of telomerase activity and TERT miRNA-mediated antioxidant effect via promotion of SOD activity	mouse model	[185,186]
Flavonoid—Genistein *Soiae semen*	Bilateral effect on telomerase activity: Reduced hTERT transcription and reduced telomerase activity in higher concentrations (50 µM), and activation of telomerase in lower concentration (0.5 − 1.0 µM)	MCF-7 cell line (human breast cancer)	[146,187]
Resveratrol *Red Grape*	Bilateral effect on telomerase activity: activation of telomerase via the upregulation of SIRT 1 in epithelial and endothelial progenitor cells and telomerase activity inhibition in cancer cells	epithelial and endothelial progenitor cells cancer cells	[188,189,190,191]
Epigallocatechin gallate *Green Tea*	Reduction of hTERT expression	cervical adenocarcinoma	[192]
Silibinin *Milk Thistle*	Reduced TERT expression and telomerase activity	LNCaP cells (human prostate carcinoma)	[193]

## Data Availability

Not applicable.

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
