# Peer review of "Potential of Naturally Derived Compounds in Telomerase and Telomere Modulation in Skin Senescence and Aging"

_ijms, 2021, doi:10.3390/ijms22126381_

Round 1
Reviewer 1 Report
Comments to the authors
The review touches a timely and interesting topic of the potential of natural compounds to modulate telomere and telomerase activities in relation to skin aging.
General comments:
- Although the study is interesting and a number of issues remain not discussed/addressed.
- In addition, a chapter addressing the expression of telomerase in skin (stem) cells and the impact of telomerase positive stem cells on skin aging should be included (e.g. telomerase transgenic mouse models). Not all cells in the skin are telomerase positive (e.g. Härle-Bachor and Boukamp, 1996; Ramirez et al., 1997); in fact, the majority of them are telomerase-negative. It would be helpful, if the authors include a figure with skin structure highlighting telomerase positive cells.
Specific comments:
- Reference 1 does not fit to the text in lines, 25-27 and line 31. There are a large body of publications, which better suit.
- Sentence in line 38-39 is obsolete. There is no information.
- Page 2: mol. Base to senescence and aging: it is important to mention that proliferative limitation of cells in culture has been shown for skin fibroblasts. Moreover, this depends on culturing condition where there is a impact of low/high oxygen conditions. The references should be added.
- Page 3, line79. Indicate that TTAGGG is conserved in vertebrates but not in all organisms
- Non-telomeric functions of telomerase: there are more recent findings and reviews on the non-telomeric functions of telomeres and telomerase, including the suppression of replication stress, DDR, gene expression (e.g. Suram et al., 2012, Meena et al., 2015; Hewitt et al., 2012; Gonzalez et al., 2014). The older reference from 2005 should be extended with original citations.
- The relevance of the polyphenols for telomeres and telomerase in the context of skin aging is missing. In fact, this is scarcely discussed for the other counpunds/moleculs as well. Instead of just listing the compounds and references, the authors should discuss their impact on telomerase/telomeres, e.g. transcriptional regulation, G4-structure, ROS generation, DDR-respoinse a.s.o.
Reviewer 2 Report
While the submitted manuscript describes an interesting application of natural ingredients on skin aging based on telomere biology, reviewer finds it rather hard to apprehend the authors' intentions. First of all, the English is quite poor and there are many in appropriate (non-scientific) expressions, typo errors, and grammatical errors as well.
Secondly, some of the references do not represent and support authors' claims and replaced.
Most of all, submitted manuscript does not properly cover the title; effects of natural ingredients on telomere or telomerase. On the contrary, the manuscript explains the general anti-aging effects of natural ingredients.
Author Response
First, we would like to thank the reviewers for the thorough revision and constructive comments. Please find below our response.
Concerning the response to reviewer 2:
Comments and Suggestions for Authors
While the submitted manuscript describes an interesting application of natural ingredients on skin aging based on telomere biology, reviewer finds it rather hard to apprehend the authors' intentions. First of all, the English is quite poor and there are many in appropriate (non-scientific) expressions, typo errors, and grammatical errors as well.
Secondly, some of the references do not represent and support authors' claims and replaced.
Most of all, submitted manuscript does not properly cover the title; effects of natural ingredients on telomere or telomerase. On the contrary, the manuscript explains the general anti-aging effects of natural ingredients.
English was amended, references were verified and a more indepth assessment of the anti-aging aspects, based on the signaling pathways, was added. Additional assessment of mechanistic pathways was enrolled, mouse model was evoked and tables were improved. All the changes were marked with yellow color.

Round 2
Reviewer 1 Report
The authors revised the manuscript satisfactorily according to my comments and suggestions.
Author Response
Dear Reviewer,
We would like to thank you for your constructive comments.
We are also glad to hear, that you have appreciated our manuscript.

Reviewer 2 Report
While the re-submitted manuscript improved much, there are still many points of revision required before being accepted.
First of all, there are still many typo errors and unclear expressions.
Also, reviewer thinks that the submitted manuscript is too long and rather focusless. It should be shortened, more focusing on telomere and telomerase in skin aging.
Author Response
Dear Reviewer,
First, we would like to thank the reviewer for constructive comments.
We thoroughly revised the typo errors and corrected the grammar.
The manuscript is now shorter and focused on the telomere/telomerase and skin aging association.
The structure was changed significantly so it was difficult to indicate precisely what were the amendments.
The new part of the text was labeled with red color.
